# Automatic Diagnosis of Pulmonary Embolism Using an Attention-guided Framework: A Large-scale Study

**Luyao Shi** [1]                                                                    luyao.shi@yale.edu

**Deepta Rajan** [2]                                                                 drajan@us.ibm.com

**Shafiq Abedin** [2]                                                                sabedin@us.ibm.com

**Manikanta Srikar Yellapragada** [3]                                                msy290@nyu.edu

**David Beymer** [2]                                                                 beymer@us.ibm.com

**Ehsan Dehghan** [2]                                                              edehgha@us.ibm.com

[1] *Department of Biomedical Engineering, Yale University, New Haven, CT, USA*

[2] *IBM Almaden Research Center, San Jose, CA, USA*

[3] *New York University, New York, NY, USA*

**Editors:** Under Review for MIDL 2020

## Abstract

Pulmonary Embolism (PE) is a life-threatening disorder associated with high mortality and morbidity. Prompt diagnosis and immediate initiation of therapeutic action is important. We explored a deep learning model to detect PE on volumetric contrast-enhanced chest CT scans using a 2-stage training strategy. First, a residual convolutional neural network (ResNet) was trained using annotated 2D images. In addition to the classification loss, an attention loss was added during training to help the network focus attention on PE. Next, a recurrent network was used to scan sequentially through the features provided by the pre-trained ResNet to detect PE. This combination allows the network to be trained using both a limited and sparse set of pixel-level annotated images and a large number of easily obtainable patient-level image-label pairs. We used 1,670 sparsely annotated studies and more than 10,000 labeled studies in our training. On a test set with 2,160 patient studies, the proposed method achieved an area under the ROC curve (AUC) of 0.812. The proposed framework is also able to provide localized attention maps that indicate possible PE lesions, which could potentially help radiologists accelerate the diagnostic process.

**Keywords:** Deep learning, computer-aided diagnosis, pulmonary embolism, attention

## 1. Introduction

Pulmonary Embolism (PE) is a sudden blockage in a pulmonary artery by a clump of material, most often a blood clot, that is usually formed in the deep veins of patients' legs and travels in the blood-stream up to the lungs. PE is a life-threatening disorder associated with high mortality and morbidity, with an estimated 100,000 deaths per year (Beckman et al., 2010). One out of four people who have a PE die without warning, and 10% to 30% of people die within one month of diagnosis (Beckman et al., 2010). Therefore, prompt diagnosis and immediate initiation of therapeutic action is crucial. Contrast-enhanced chest CT is commonly used for PE diagnosis. However, manual reading of all the CT slices by radiologists is laborious, time consuming and often complicated by false positives caused by various PE look-alike image artifacts, lymph nodes, and vascular bifurcation, among many

others (Liang and Bi, 2007). Moreover, the accuracy and efficiency of interpreting such a large image data set is also limited by human's attention span and eye fatigue.

Recent advancements in deep learning have enabled computer-aided diagnosis (CAD) algorithms to provide accelerated diagnosis that can assist medical professionals in a variety of medical abnormality detection tasks (Tajbakhsh et al., 2015; Gondal et al., 2017; Braman et al., 2018; Guan et al., 2018; Huang et al., 2019; Rajan et al., 2019), including diabetic retinopathy (Gondal et al., 2017), emphysema (Braman et al., 2018) and PE (Tajbakhsh et al., 2015; Huang et al., 2019; Rajan et al., 2019). One enduring challenge of training a deep neural network using medical imaging data is the difficulty to collect a sufficiently large annotated data set. Many approaches thus focus on utilizing the more abundant and easily obtainable report-based labeled data to train the networks (Gondal et al., 2017; Braman et al., 2018; Guan et al., 2018). In this approach, instead of relying on manually produced pixel-level annotations, patient-level labels are extracted from radiology reports that accompany images.

In order to help the user understand how and why the network makes predictions, one can also obtain attention maps (Zhou et al., 2016; Selvaraju et al., 2017) for a given input image with back-propagation on a convolutional neural network (CNN), which reveals which regions on the input image contribute to the final prediction. Attention maps can also be used to provide localization information and help build confidence in the network predictions. However, supervised by classification loss only, such an end-to-end training often results in attention maps that only cover the most discriminative regions, but not necessarily the regions that contain the desired objects (lesions in medical applications) for classification. For instance we may encounter a bias in the training data where PE lesions incidentally always correlate with the same background regions (for example ribs or vertebrae), in this case the training has no incentive to focus attention only on the PE. In some worse cases the training might be distracted and only focuses on those background regions when the tiny PE lesions are hard to detect. We have observed this kind of distracted attention in our study (see Figure 3) as well as in the literature (Huang et al., 2019). The generalization ability of the trained model is likely to degrade when the testing data has a different correlation. Providing supervision on the network attention is expected to improve the network performance (Li et al., 2018).

In this work, we first trained a 2D slice-level classification network with attention supervision using a relatively small data set of pixel-level annotated slices. Then, we trained a recurrent network to scan through the features provided by the slice-level classifier, taking into account the spatial context between the 2D images, and produce a patient-level PE prediction. For this training, we used a large data set of label-only volumetric images. We show that attention training (AT) provides much better results on the PE detection task compared with training using classification loss only. We also show that using a large data set of volumetric images without pixel-level annotations can improve classification results even for small objects like PE.

The proposed framework achieved improved results, in terms of AUC, when compared with the state-of-the-art (Huang et al., 2019) despite being tested on a much larger and more diverse testing set. Our method can also provide localized attention maps that indicate possible PE lesions, which could potentially help radiologist accelerate the diagnostic process. Compared with the previous works that rely solely on pixel-level annotations or

slice-level labels (Tajbakhsh et al., 2015; Huang et al., 2019) for training, our method can take advantage of large data sets of easily obtainable image-label pairs.

## 2. METHOD

Our proposed framework consists of two stages. In stage I, a 2D convolutional network is trained on a limited set of pixel-level annotated image slices. The patient-level PE prediction is obtained in stage II. For each slice in a volumetric CT image, the network from stage I serves as an image encoder and provides encoded features for stage II. A recurrent network in stage II incorporates the features from all the slices and finally provides the PE prediction.

### 2.1. Stage I: Attention-guided Network

In the first stage, a classification network is trained as an image encoder based on annotated 2D image slices. To improve the overall performance of the encoder network, we used a similar approach to the extension of guided attention inference networks (GAIN$_{ext}$) (Li et al., 2018), which supervises the attention maps while training the network. In this way, the network prediction is based on the suspicious PE regions on which we expect the network to focus. This is achieved by training the network with a combination of classification and attention losses.

Attention maps were often used as a retrospective network visualization method (Zhou et al., 2016; Selvaraju et al., 2017; Gondal et al., 2017; Guan et al., 2018). In order to make the attention trainable, we must generate the attention maps during training. Based on the fundamental framework of Grad-CAM (Selvaraju et al., 2017), for a given image $I$, let the last convolutional layer produce $K$ feature maps, $f^k \in \mathbb{R}^{u \times v}$, we first compute the gradient of the score for class $c$, $g^c$, with respect to the feature maps $f^k$, and obtain $\partial g^c / \partial f^k$. The neuron importance weights $\alpha_k^c$ are obtained by global average-pooling over these gradients flowing back.

$$\alpha_k^c = \frac{1}{u \times v} \sum_i \sum_j \frac{\partial g^c}{\partial f_{ij}^k} \tag{1}$$

These weights capture the importance of feature map $k$ for the target class c. We then calculate the weighted combination of the activation maps, followed by a ReLU operation to obtain the attention map $A^c$ for class $c$:

$$A^c = ReLU(\sum_k \alpha_k^c f^k) \tag{2}$$

ReLU was applied because we are only interested in the features that have a positive influence on the class of interest. The attention map is then normalized by its maximum value to range between 0 and 1, if it contains non-zero values. For negative samples, the attention maps were most likely to be all zeroes, and normalization was not performed. The spatial resolution of the attention map is the same as the final convolutional layer. In this work, we used ResNet18 (He et al., 2016) as the classification network, thus the attention map contains $24 \times 24$ pixels for an input image of size $384 \times 384$. We also downsampled the pixel-level PE segmentation masks by the same factor to match the size of the attention

maps. Bi-linear interpolation was used for downsampling, thus the resulted low-resolution mask was a blurred version of the original mask and contained floating numbers between 0 and 1. We produced a binary map from the low-resolution masks by setting any values larger than 0 to 1. Note that this will result in slightly larger PE masks than their actual sizes, though the task here is supervising the attention instead of training a segmentation network. As long as the the masks cover the PE lesions, they do not necessarily have to be exact.

In this application there are only two classes: PE-positive and PE-negative. Let $y$ be the ground-truth PE classification label and $\hat{y}$ be the network PE prediction score; $A$ be the attention map corresponding to the positive class and $M$ be the down-sampled PE segmentation mask. Our loss function used to update the network parameters is defined as:

$$L_{total} = L_{CE}(\hat{y}, y) + \lambda L_{CDC}(A, M) \tag{3}$$

where the classification loss, $L_{CE}$, is the categorical cross-entropy loss, and the attention loss, $L_{CDC}$, is the continuous dice coefficient (Shamir et al., 2019). $\lambda$ is the weighting parameter depending how much emphasis we place on the attention supervision (we used $\lambda = 1.0$ in our experiments). For the negative samples, an all-zero mask with the same size was used. We also used slab-based input images (5 slices/slab in this study) to better utilize the neighboring spatial information, where only the central slice in a slab has the corresponding annotation mask. An illustration of the network training in Stage I is shown in Figure 1.

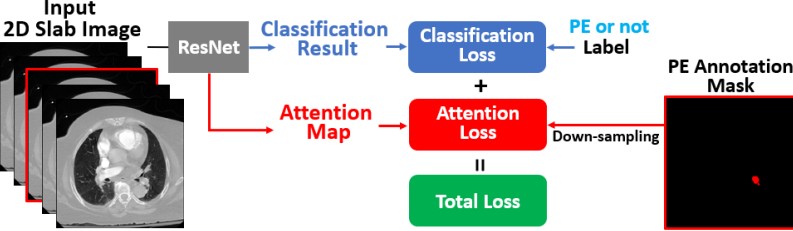

Figure 1: Network training in Stage I.

## 2.2. Stage II: Recurrent Framework

The ResNet in stage I can provide PE inference on each 2D image slice. In order to obtain patient-level PE inference, the results from different slices need to be integrated. A simple and naive approach is to summarize (average or max) these slice-level predictions into a single bag probability. However, this approach fails to account for spatial context between the slices in a volumetric image. For example, isolated positive samples throughout a volume are more likely to represent false positives due to noise, while a series of consecutive positive slices may indicate a positive PE on the patient level. In our experiments, this assembling approach only resulted in slightly better results than random guessing.

Alternatively, recurrent neural networks such as convolutional long-short term memory (Conv-LSTM) (Xingjian et al., 2015) are capable of interpreting and summarizing patterns

among correlated samples. Braman et al. used bidirectional Conv-LSTM units to scan through a series of consecutive slices from an image volume to detect emphysema and obtained better results compared with 3D CNN and multiple instance learning (Braman et al., 2018).

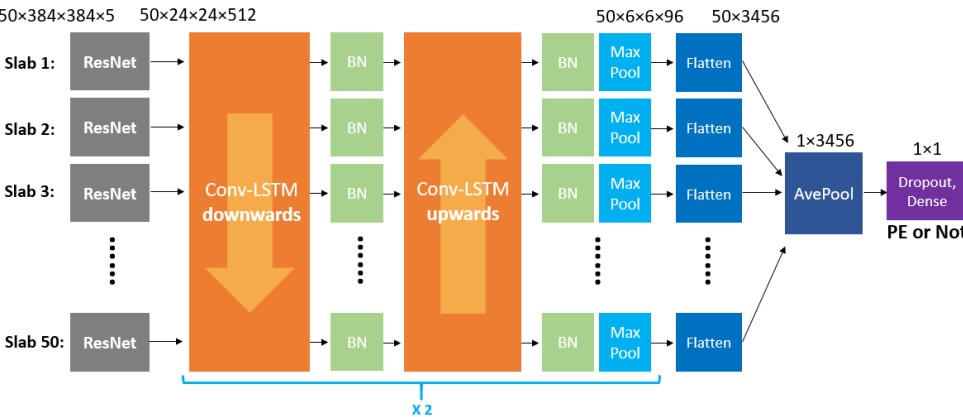

Figure 2: Recurrent network in Stage II.

Our architecture is depicted in Figure 2. We first applied the pre-trained ResNet18 from stage I to each slab in a volumetric image. Instead of using the classification or attention map of the ResNet as the input to the recurrent network, we used the last convolution layer after activation to preserve the complete information embedded in the feature maps. The proposed recurrent framework contains 2 units. Each unit consists of one ascending and one descending Conv-LSTM units, followed by a max-pooling layer with pooling size of 2×2 and stride of 2. Batch normalization was applied after each Conv-LSTM layer. Each Conv-LSTM layer has 96 filters with a 3×3 kernel size. Channel-wise dropout of rate 0.2 was also applied to the inputs and the recurrent states in the Conv-LSTM layers. The final Conv-LSTM layer outputs a series of features, which summarizes the network findings after processing through the image volume. After an average pooling layer to integrate the features along the z-axis, a fully connected layer with sigmoid activation and dropout with a rate of 0.5 computes probability of presence of PE. Binary cross entropy (BCE) loss was used to update the network.

## 3. EXPERIMENTS AND RESULTS

### 3.1. Data

All data used in our algorithm training and evaluation was private data obtained from our collaborative partners. All private data used were anonymized. HIPPA was fully enforced and all data were handled according to the Declaration of Helsinki. Our data came from various hospitals and were acquired from various makes and models of CT scanners. All studies are contrast-enhanced, but are not limited to PE protocol studies. For training and validation, we used 5,856 studies marked as positive by an NLP algorithm used by the data provider. We also received a large cohort of contrast-enhanced CT study and

radiology report pairs from our data provider without any labels. From this cohort, we selected 5,196 studies as PE negatives decided by an in-house NLP algorithm. Furthermore, a subset of positive and negative studies were more rigorously verified for training and validation in stage I. We had 1,670 positive studies manually annotated by board-certified radiologists. For each annotation, we asked a radiologist to segment every embolism in slices approximately 10 mm apart. This annotation process leaves several un-annoated slices between two annotated slices. In the end, 10,388 slices were annotated.

Additionally, we selected another 2,160 independent studies (517 positive and 1,643 negatives) as our test set. Radiology reports of the studies in the test set were manually reviewed to confirm the label. Note that, due to the specific anonymization protocol used by our data provider, we are unable to determine if two studies belong to the same patient.

### 3.2. Network training

To train the network in stage I, we used image slabs produced from 10,388 annotated slices as positive samples and an equal number of randomly selected slabs from 593 negative studies as negative samples. These studies have various slice thicknesses ranging from 0.5 mm to 5 mm, with a median of 2 mm. This could cause instability when we train the network using slab-based data. Therefore, we first resampled the volumetric images to have a 2.5 mm slice thickness using bilinear interpolation. Then, image slabs of 5 slices were selected around the annotated slices . All the image slabs were cropped to size $384 \times 384$ around the center, and the values between -1024 HU and 500 HU were mapped to $0 - 255$.

Labeled volumetric images were used for training of stage II and for testing. For stage II training, in addition to the studies used in stage I, we further included 4,186 positive and 4,603 negative studies, which resulted in a total of 5,856 positive and 5,196 negative volumetric images. The image slices containing lung were identified using an in-house lung segmentation tool. Those slices were resized to 200 slices, and then sampled every 4 slices to obtain 50 slabs where each slab contains 5 slices. After image cropping and value mapping (same as in stage I), the input image size of each study was $50 \times 384 \times 384 \times 5$. For each study, the input to the network in stage II is the output of last convolutional layer of the trained ResNet in stage I, which is a tensor of size $50 \times 24 \times 24 \times 512$. During the training of the stage II network, the weights of the stage I network were fixed and not updated.

The framework was implemented using Keras and trained on two NVIDIA Tesla P100 GPUs. The ResNet in stage I was trained for 100 epochs with a batch size of 48 and the model with the highest validation accuracy was selected and used in stage II. The recurrent network in stage II was trained for 50 epochs with a batch size of 32 and the model with the highest validation accuracy was finally selected. For both stages, 80% of the data was used for training and the remaining 20% for validation. We used the Adam optimizer with a learning rate of $10^{-4}$.

### 3.3. Model Comparison in Stage I

Figure 3 shows an example of the generated attention maps of an image slab in the validation data set with and without attention training in stage I. The attention maps with attention training are more localized and show strong concordance with the annotation masks produced by the radiologist. On the other hand, the attention maps without atten-

tion training are widely distributed and mostly focused on irrelevant regions where no PE is present. The ROC curves in Figure 4 (left) show the effect of using attention training on slice-level classification performance by increasing the AUC to 0.953 from 0.932 on the validation set.

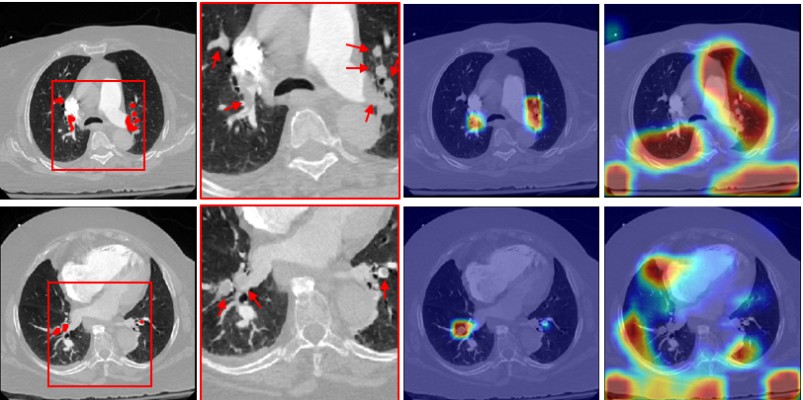

Figure 3: Examples of generated attention maps for two validation image slabs with and without attention training in stage I. First column, images with annotation; Second column, zoomed regions with PE; Third column, attention maps with attention training; Fourth column, attention maps without attention training.

### 3.4. Test Set Evaluation

We measured the performance of our overall pipeline in patient-level prediction of PE on a test set with 2,160 studies in 3 scenarios. In all these scenarios, the first stage network was trained using annotated positive studies (1,670 cases) and 593 negative studies. Also, the first stage network was frozen during the training of the second stage. In the first scenario, we only used the data that were used to train the first stage to train the second stage as well (no label-only studies). The first stage was trained with a combination of classification and attention losses as in equation (3). In the second and third scenarios, in addition to the data that were used in the first scenario, we used 8,789 label-only studies to train stage II. The difference between the second and third scenarios is in the loss function that was used to train stage I. In the second scenario, the stage I network was trained using classification loss only. In the third scenario, similar to the first scenario, we used a combination of attention and classification loss.

Figure 4 (right) shows the ROC of our pipeline for the three test scenarios. As can be seen, using the attention loss significantly improved our pipeline performance by increasing the AUC from 0.643 (red curve) to 0.812 (blue curve) when trained on the same data. This implies that the ResNet in stage I obtained better PE feature extraction ability with attention training, since the output of the ResNet's last convolutional layer was used as the input in the recurrent network in stage II (neither the classification results nor the attention maps were used). The low performance of the pipeline without attention loss shows that a

network with distracted attention that focuses on the correlated background features might still obtain high accuracy in 2D image classification, as can be seen in Figure 4 (left), but will not be able to generalize well on patient-level classification based on the inconsistent feature patterns.

Figure 4 (right) also demonstrates that our pipeline can use a large amount of easily obtainable label-only data samples to increase its accuracy. Our AUC increased from 0.739 (green curve) to 0.812 (blue curve) when we added 8,789 label-only studies to our training data. This is important for scalability of a pipeline to be trained on a large data set. Relying solely on fine annotated data sets for training is not feasible when a large data set is required.

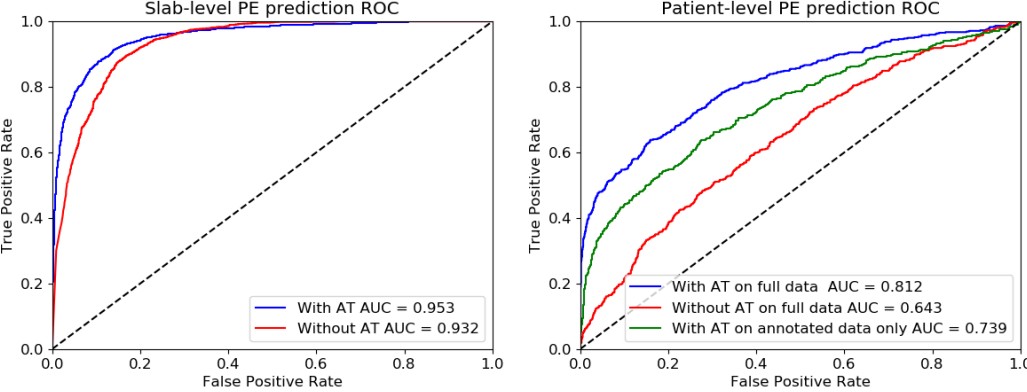

Figure 4: Left: ROC curves of the ResNet's slab-level PE prediction result on the validation data in stage I. Right: ROC curves of the whole framework's patient-level PE prediction on the 2160 testing data set.

### 3.5. Comparison with State-of-the-art

Compared to the state-of-the-art PENet (Huang et al., 2019), which achieved an AUC of 0.79 and accuracy of 0.74 on an internal test set, and AUC of 0.77 and accuracy of 0.67 on an external test set, our method obtained an AUC of 0.812 with confidence interval [0.789, 0.835] (Sun and Xu, 2014) and accuracy of 0.781 (threshold of 0.5 was used for both PENet and our method) on a much larger test set (2160 versus approx. 200). Moreover, the studies they used were acquired under the same PE protocol at the same institute with a consistent high resolution slice thickness (1.25 mm), whereas our data was acquired from various hospitals under different imaging protocols so the images had different noise levels and slice thickness (0.5 mm - 5 mm).

Our proposed model was also compared with a 3D CNN model that has demonstrated success in acute aortic syndrome detection (Yellapragada et al., 2020). The model starts with an I3D model (3D CNN pretrained on video action recognition dataset) (Carreira and Zisserman, 2017), followed by Conv-LSTM layers and dense layers as the classifier. The model was trained only on our patient-level labeled data, and resulted in an AUC of 0.787

and accuracy of 0.727 (used threshold of 0.5) on our test set, which is still inferior to our result. A summary of the test results for different methods is shown in Table 1.

Table 1: Test results for different methods. For PENet, the results on both their internal (int.) and external (ext.) test sets are given.

| Approach | Testset Size | AUC | Accuracy |
|---|---|---|---|
| PENet (int.) | 198 | 0.79 | 0.74 |
| PENet (ext.) | 227 | 0.77 | 0.67 |
| 3D CNN | 2160 | 0.787 | 0.727 |
| Proposed | 2160 | 0.812 | 0.781 |

### 3.6. PE visual localization

Figure 5 shows examples of the attention maps of two positive PE studies from the test set. The highlighted attention-focused regions accurately identify the PE lesions, which could potentially serve as a quick and convenient tool to help radiologists localize PE lesions.

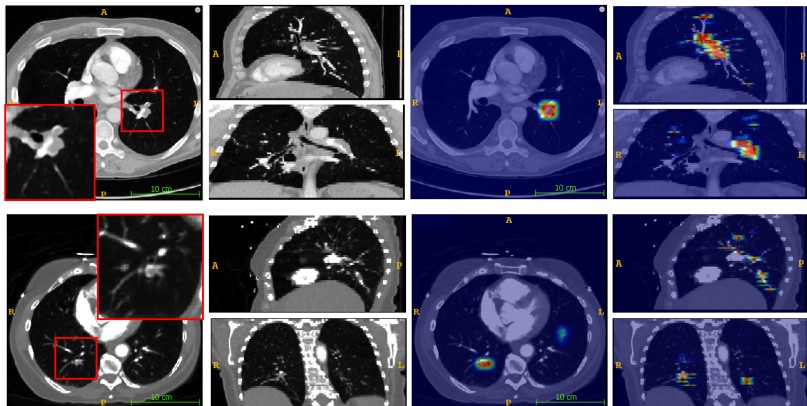

Figure 5: Two examples of the attention maps which can provide PE visual localization.

## 4. CONCLUSIONS AND FUTURE WORK

We presented a deep learning approach that is capable of detecting PE on contrast-enhanced CT images which can be trained using a combination of pixel-level annotated and patient-level labeled data. Our framework consists of two stages. Stage I is a slab-based classification network trained with an attention loss that predicts the presence of PE in a image slab, and stage II is a recurrent network that takes into account the spatial context between slabs in a volumetric image and provides final patient-level PE prediction. We showed that attention training can provide better PE feature extraction ability for the network that

processes slab-based images in stage I. The proposed framework achieved improved results compared to the state-of-the-art on a much larger and more complex testing set. To the best of our knowledge, our evaluation involves the largest number of patient studies among all the research studies on automatic PE detection. The proposed framework also outperformed a 3D CNN network on the same test set. One benefit of the proposed framework is its ability to provide localized attention maps that indicate possible PE lesions, which could potentially help radiologists accelerate the diagnostic process. Also, compared to prior art, our pipeline can learn from a combination of annotated and labeled data sets that is very useful in training on large-scale data sets or transfer learning. Beyond PE, this approach is also applicable to other disease/abnormality detection problems with the availability of the combination of pixel-level annotated data and patient-level volumetric imaging data with binary labels.

In this work, the first stage weights were not updated during the training of the second stage. End-to-end training may improve the results as a large cohort of label-only data may help stage I network as well. In stage I, we used ResNet18 instead of deeper networks to balance between performance and our GPU resources, although using more efficient network structures, for example, DenseNet (Huang et al., 2017), can be explored in the future. Additionally, optimizing the $\lambda$ parameter and extending our framework to higher dimensions (van de Leemput et al., 2019) can also be explored in the future. One limitation of our validation study is that, due to the specific anonymization process used by our data provider, we do not know whether two studies belong to the same patient. Nonetheless, our data were collected from a large number of hospitals and over several months of time. We believe the chance of leakage between training and test set is very small. Future work will include validation of our pipeline on a new and completely independent test set. Another limitation is that, to reduce the training time and the required GPU memory, we resampled all the images to have a slice thickness of 2.5 mm, which might be too thick for detecting sub-segmental PEs. However, these small embolisms are less actionable and less dangerous than the larger ones. In addition, we are interested in analyzing all contrast-enhanced CT images, which usually have a slice thickness larger than 2.5 mm, instead of analyzing only CT pulmonary angiography (CTPA) ones that usually have a small slice spacing. Nonetheless, the model can be re-trained with smaller slice thicknesses without any change to the architecture or methodology. One strength of our pipeline is that it can be trained using volumetric images with only binary labels. As a result, it is well suited for transfer learning using new data sets from different hospitals. Evaluation of our pipeline in such a scenario is also part of the future work.

## Acknowledgments

We would like to thank Yiting Xie, Benedikt Graf and Arkadiusz Sitek from IBM Watson Health Imaging for helping us generate the 3D CNN results.

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
