# OpenReview forum: "Automatic Diagnosis of Pulmonary Embolism Using an Attention-guided Framework: A Large-scale Study"
_MIDL.io/2020/Conference — MIDL 2020_

### Official Review · AnonReviewer1 · 2020-03-11
**Detecting pulmonary embolism on chest CTs**

**Rating:** 3
**Confidence:** 5
**Recommendation:** Poster

**Summary:**

The paper describes a method to detect pulmory embolism (PE) and indicate possible PE lesions. The method was validated on a large internal dataset with a total of more than 10,000 studies for training and around 2,000 studies for testing. The proposed method consists of two main steps; step 1 focuses on generating per slice features trained on attention masks and classification labels while step 2 focuses on using the features of each slice to predict patient-level PE.

**Strengths:**

- The idea is simple and easy to follow; I particularly like the idea in stage 2 to move from slice level PE to patient level PE inference
- The paper is well written
- Validated on a large dataset (10,000+ training, 2,000+ testing)
- State-of-the-art results

**Weaknesses:**

- The only major weakness is the lack of novelty
- Minor weaknesses: Is there a possibility of a training/test leak? In section 3.1 it was mentioned that "Note that, due to the specificc anonymization protocol used by our data provider, we are unable to determine if two studies belong to the same patient.".

**Justification Of Rating:**

The paper is sound and easy to follow and was validated properly; large dataset for validation. It is well written and ideas were clearly explained and validated with proper experiments. The only gripe is the lack of novel ideas.

**Paper Type:**

both

**Special Issue:**

no

---

> ### Author Response · Authors · 2020-03-27
> **Response to reviewer 1**
>
> We thank the reviewer for the thorough review and helpful suggestions! We will try to address the questions and suggestions below:
> Comments: Is there a possibility of a training/test leak? In section 3.1 it was mentioned that "Note that, due to the specific anonymization protocol used by our data provider, we are unable to determine if two studies belong to the same patient.".
> Response: We share the reviewer’s concern regarding the training/test leak of the data. Unfortunately, the names and the number of hospitals involved are not known to us. However, we do know that our data partner collected the data from a large number of hospitals and over several months of time. We believe the chance of leakage between training and test set is small.

---

### Official Review · AnonReviewer3 · 2020-03-12
**Simple but valuable method in PE diagnosis with explainable features**

**Rating:** 3
**Confidence:** 4
**Recommendation:** Oral

**Summary:**

In this paper, the authors propose a two-stage method to diagnose pulmonary embolism (PE) in CT volumes. In stage I, PE masks are utilized to supervise the learning of attention maps during the training of a 2D classification network. In stage II, the trained network is then used to extract meaningful features for each slice, and those features are passed to a recurrent neural network for the final diagnosis of PE. Experiments on a large dataset show the effectiveness of their method in PE diagnosis. What's more important is that the diagnosis process can be visualized rather than a black box.

**Strengths:**

1. The proposed method can provide explainable features (PE attention maps) about the decision process of deep learning in PE diagnosis. It can help the doctor to understand why the method gives the current result, thus it is more acceptable to doctors compared to a regular classification model.
2. The method is trained and validated on a large-scale dataset, and is shown to be effective in PE diagnosis.
3. The paper is overall clear and easy to follow.

**Weaknesses:**

1. The comparison to the state-of-the-art method is not reasonable, because the two methods use different training and test data. The performance of a method may vary a lot on different datasets.
2. The downsampling of PE masks from 384x384 to 24x24 may greatly affect the actual sizes of PEs, especially for small PEs. For example, it seems that the attention score for the right small PE in the second row of Figure 3 is not good enough.
3. The description of the three scenarios in section 3.4 is not clear enough. Can be improved by summarizing them in a table.

**Detailed Comments:**

1. I am curious about the accuracy of the attention maps in the test set, i.e., what is the percentage of PE lesions the attention maps can correctly identify?
2. About stage-2, it will be better to provide the performance using 3D CNN or other 2D methods (e.g., concatenate the features of each 2D slices and send them into a 2D model).

**Justification Of Rating:**

The interpretability of this method is very important to the development of AI methods in medical image analysis, especially for applying novel methods to practical medical scenarios. However, there are some issues need to be addressed. I will change to strong accept if the authors can answer my concerns.

**Paper Type:**

methodological development

**Questions To Address In The Rebuttal:**

1. Report the confidence interval of the AUC score.
2. Why not upsample the attention map and compute the attention loss with the masks in the original size?

**Special Issue:**

no

---

> ### Author Response · Authors · 2020-03-27
> **Response to reviewer 3**
>
> We thank the reviewer for the thorough review and helpful suggestions! We will try to address the questions and suggestions below:
> Comment 1: The comparison to the state-of-the-art method is not reasonable, because the two methods use different training and test data. The performance of a method may vary a lot on different datasets.
> Response 1: The training of PENet requires the training data to be labeled on a slice level for the presence/absence of a PE. Currently, we do not have such annotated data. For each annotation, we asked the radiologist to segment every embolism in slices approximately 10 mm apart. This annotation process leaves several un-annoated slices between two annotated slices. Therefore, it is currently not feasible to implement PENet and evaluate it based on our data.
> We do agree that comparing a method with existing methods on the same dataset is essential for fair evaluation of the method. However, we did not find many published works on patient-level PE diagnosis, and the existing works require training datasets with specific labeling conventions that we do not currently have access to. Providing a public test set available in the future, a fair comparison can be performed. We will address this limitation in the discussion section in the revised manuscript.
>
> Comment 2: The downsampling of PE masks from 384x384 to 24x24 may greatly affect the actual sizes of PEs, especially for small PEs. For example, it seems that the attention score for the right small PE in the second row of Figure 3 is not good enough. Why not upsample the attention map and compute the attention loss with the masks in the original size?
> Response 2: We want to clarify that our goal is not trying to train a PE segmentation network. The attention map is just a visualization tool to reveal the network’s attention in a coarse level due to its much smaller size in image space. The complete information that contributes to the final prediction lies in the last convolutional layer’s output, which is what we used as input to State II. Therefore, we believe that as long as the downsampled masks cover the PE lesions, the attention training can enforce the network to focus on the correct attention. Although we agree that upsampling the attention map and computing the attention loss with the masks in the original size is an alternative way and is worth investigating in the future.
>
> Comment 3: The description of the three scenarios in section 3.4 is not clear enough. Can be improved by summarizing them in a table.
> Response 3: We thank the reviewer for the helpful suggestion. We can summarize them in a table if space permits.
>
> Comment 4: Report the confidence interval of the AUC score.
> Response 4: [0.789, 0.835] using DeLong’s method [1, 2].
> [1] E. DeLong, D. DeLong, D. Clarke-Pearson, "Comparing the areas under two or more correlated receiver operating characteristic curves: A nonparametric approach", Biometrics, pp. 837-845, 1988.
> [2] X. Sun and W. Xu, "Fast Implementation of DeLong’s Algorithm for Comparing the Areas Under Correlated Receiver Operating Characteristic Curves," in IEEE Signal Processing Letters, vol. 21, no. 11, pp. 1389-1393, Nov. 2014.
>
> Comment 5: 1. I am curious about the accuracy of the attention maps in the test set, i.e., what is the percentage of PE lesions the attention maps can correctly identify? 2. About stage-2, it will be better to provide the performance using 3D CNN or other 2D methods (e.g., concatenate the features of each 2D slices and send them into a 2D model).
> Response 5: For the 2160 test set that we used, we only have the patient-level labels but we do not have the annotations for each PE lesion, therefore it is not currently possible to find out the percentage of PE lesions the attention maps can correctly identify. However, as we continue our research in this field, we will evaluate this in the future once we have the annotations.
> We did not compare with a 3D CNN approach because a recent study has shown that using bidirectional Conv-LSTM to scan through a series of consecutive slices of an imaging volume and output as a final set of features characterizing overall disease presence has achieved outperformed a 3D CNN and multiple instance learning (Braman et al., 2018) in detection of emphysema. As PE is actually much smaller than the emphysematous portion of a lung, we expect the advantage of using bidirectional Conv-LSTM over 3D CNN will be even larger.

---

> > ### Comment · AnonReviewer3 · 2020-04-02
> > **The comparison to the baseline method is still no good**
> >
> > Although the authors explained the reason why they made a comparison using different datasets, it is still not good to do so. Because of this, I suggest adding the results of 3D CNN for another baseline method comparison.

---

> > > ### Author Response · Authors · 2020-04-03
> > > **Adding the result of 3D CNN for another baseline method comparison**
> > >
> > > To compare to a baseline model with 3D CNN, we composed a model that starts with an I3D model (3D CNN pretrained on action recognition) followed by Conv LSTM layers and dense layers as the classifier. This network structure has demonstrated success in acute aortic syndrome detection, in a soon to be published paper.
> > >
> > > The network was trained only on the patient-labeled data, and resulted in an AUC of 0.787 on our test set, which is not as good as the AUC of 0.812 by our proposed method.
> > >
> > > We will add details about the baseline model and comparison to our model in the final paper.

---

### Official Review · AnonReviewer2 · 2020-03-13
**Automated classification of PE in CT images using a deep learning framework with attention supervision**

**Rating:** 4
**Confidence:** 5
**Recommendation:** Oral

**Summary:**

This is a well written paper which used a two stage deep learning framework to classify contrast-enhanced CT images for the presence of PE. A 2D first stage ResNet is trained to classify slices with attention supervision. There are two loss terms: one for the classification error (categorical cross-entropy) and one for the attention (continuous dice coefficient). The second stage is a recurrent neural net based on bidirectional convolutional LSTM using the features from the last layer of the ResNet. The second stage is trained with extra cases for which only a scan level label is available. The method is validated and tested on a large dataset coming from multiple sources and shows good performance.

**Strengths:**

- This is a well written paper with a clear introduction into the problem at hand
- The paper provides a framework that can be used for other applications and is explained well
- Large data set used for training, validation and testing
- Experiments show the effect of adding extra cases with no slice-level labels but only scan-level labels.


**Weaknesses:**

- There are some important details missing about the data. From how many hospitals did the data originate? What were the CT characteristics, such as the distribution of the slice thicknesses? The limitation that the authors did not know whether two studies belong to the same patient is risky. Do the authors know from how many patients these scans are? That would help to estimate the risk that there is leakage between train and test.
- No comparison about human performance and corresponding inter- and intra-observer variability
- Hard to compare with the other approaches because different dataset used, but I cannot blame the authors for this.

**Detailed Comments:**

Do the authors expect that there is a lot to gain if the lambda parameter would be optimized? It seems it is now set to 1 and now optimzed at all.

I think this paper uses a similar methodology to process 4D data using a 3D conv and LSTM on top: https://ieeexplore.ieee.org/document/8822732. If the authors agree, it would be nice to add to the discussion that this can also be extended to more dimensions.

**Justification Of Rating:**

This is a well written paper which uses a novel methodology to predict PE presence at the scan level. In addition to this, it uses a large scale dataset to train and validate the performance of the deep learning system. The method is well explained and the experiments support the claim of the paper.

**Paper Type:**

both

**Questions To Address In The Rebuttal:**

Please provide more details about the CT characteristics of the data.
Adding the patient level AUC by taking the maximum probablity from stage I would be a nice addition to the paper.
- Could it be that the network without attention just needs a lot more time for training, and was not finished with training yet (you used a fixed number of epochs)? Loss curves would have helped to analyze this. A result of this study would then be that attention supervision accelerates training.

**Special Issue:**

yes

---

> ### Author Response · Authors · 2020-03-27
> **Response to reviewer 2**
>
> We thank the reviewer for the thorough review and helpful suggestions! We will try to address the questions and suggestions below:
> Comment 1: From how many hospitals did the data originate? What were the CT characteristics, such as the distribution of the slice thicknesses? The limitation that the authors did not know whether two studies belong to the same patient is risky. Do the authors know from how many patients these scans are? That would help to estimate the risk that there is leakage between train and test.
> Response 1: We share the reviewer’s concern regarding the characteristics of our data. While some of them, such as slice thickness can be measured and will be added to the final version, the names and the number of hospitals involved are not known to us. However, we do know that our data partner collected the data from a large number of hospitals and over several months of time. We believe the chance of leakage between training and test set is small.
>
> Comment 2: No comparison about human performance and corresponding inter- and intra-observer variability.
> Response 2: Although a comparison with human performance is very interesting, we believe it is beyond the scope of this paper. However, as we continue our research in this field, we will compare the performance of our algorithm against radiologists in future.
>
> Comment 3: Adding the patient level AUC by taking the maximum probability from stage I would be a nice addition to the paper.
> Response 3: In our early investigations, we tried taking the maximum probability and mean probability from stage I on a smaller test set with 712 (385 positive, 327 negative) studies, and only obtained AUCs of 0.56 and 0.52, respectively, we thus did not further try this on the current test set with more studies.
>
> Comment 4: Could it be that the network without attention just needs a lot more time for training, and was not finished with training yet (you used a fixed number of epochs)? Loss curves would have helped to analyze this.
> Response 4: we checked the training loss curve for the network without attention training, and the loss stopped decreasing after around 50/100 epochs. We can add the loss curve in the final version of the manuscript if space permits.
>
> Comment 5: Do the authors expect that there is a lot to gain if the lambda parameter would be optimized? It seems it is now set to 1 and now optimized at all. It would be nice to add to the discussion that this can also be extended to more dimensions.
> Response 5: We thank the reviewer for the helpful suggestion. We think optimizing the lambda value might be able to bring slight improvement, but we do not expect a lot of gain. This is surely worth investigating in the future. We will add in the discussion that optimizing the lambda parameter will be included in the future study, and the proposed method can be extended to more dimensions.

---

### Official Review · AnonReviewer4 · 2020-03-16
**Automatic Diagnosis of Pulmonary Embolism Using an Attention-guided Framework: A Large-scale Study**

**Rating:** 2
**Confidence:** 3

**Summary:**

The paper proposes a two-stage patient-level PE prediction pipeline. Stage 1 learns to predict PE from 2D image slices using both slab-level and pixel-level annotated dataset. By aggregating the output of Stage 1 using the bidirectional Conv-LSTM layer, Stage 2 learns to predict PE at the patient-level. Experimental results for both slab-level and patient-level PE prediction are provided. The ablation study shows pixel-level annotation data is important for a major improvement in both slab-level and patient-level PE prediction.

**Strengths:**

Automatic Diagnosis of Pulmonary Embolism Using an Attention-guided Framework: A Large-scale Study: The paper is nicely written, and the illustrations are well-done.  In general, the paper is easy to understand. The authors have a large dataset, and clinical motivation is very clear.

**Weaknesses:**

1. Comparisons with other state-of-the-art methods are very weak and not fair. The authors compare the proposed method with only one method PENet (Huang et al., 2019), which has been trained on a different and much smaller dataset. In this scenario, it is difficult to judge whether the claimed performance improvement of the proposed method is due to a larger dataset or better method. In addition, no comparison with other known methods as cited. To have a fair comparison, the baseline methods must be evaluated on the same dataset and data-split.

2. The proposed method requires much expensive pixel-level annotation to predict patient-level annotation. It is mentioned in the paper that “[o]ne strength of our pipeline is that it can be trained using volumetric images with only binary labels” which not a methodological advantage for the proposed method only. A 3D CNN can also be trained with binary patient-level annotation only. Since the authors have a large-scale dataset, a much better approach could be training 3D CNN using patient-level annotation only and compare it, as a baseline, with the proposed method.

3. It is mentioned in the paper that “[t]he attention map is then normalized by its maximum value to range between 0 and 1”. This normalization should make the attention map similar for both negative and positive samples which should hurt the final prediction.

4. Using 2.5mm as slice thickness is probably too thick for detecting subsegment PE.


**Detailed Comments:**

N/A

**Justification Of Rating:**

1. Comparisons with other state-of-the-art methods are very weak and not fair.

2. The proposed method requires much expensive pixel-level annotation to predict patient-level annotation.

3. Using 2.5mm as slice thickness is probably too thick for detecting subsegment PE.

Please see the details of the weaknesses.


**Paper Type:**

both

**Questions To Address In The Rebuttal:**

Please see the weaknesses.

**Special Issue:**

no

---

> ### Author Response · Authors · 2020-03-27
> **Response to reviewer 4**
>
> We thank the reviewer for the thorough review and helpful suggestions! We will try to address the questions and suggestions below:
> Comment 1: Comparisons with other state-of-the-art methods are very weak and not fair. To have a fair comparison, the baseline methods must be evaluated on the same dataset and data-split. In addition, no comparison with other known methods as cited.
> Response 1: The training of PENet requires the training data to be labeled on a slice level for the presence/absence of a PE. Currently, we do not have such annotated data. For each annotation, we asked the radiologist to segment every embolism in slices approximately 10 mm apart. This annotation process leaves several un-annotated slices between two annotated slices. Therefore, it is currently not feasible to implement PENet and evaluate it based on our data.
> For the other two PE detection methods cited, [Tajbakhsh, et al., 2015] focused on the detection of individual PE instances instead of patient-level PE diagnosis and requires training data with complete PE labeling.[Rajan, et al., 2019] evaluated their method on a much smaller data set (512 studies) and obtained a lower AUC of 0.78 compared to our work.
> We do agree that comparing a method with existing methods on the same dataset is essential for fair evaluation of the method. However, we did not find many published works on patient-level PE diagnosis, and the existing works require training datasets with specific labeling conventions that we do not currently have access to. Providing a public test set available in the future, a fair comparison can be performed. We will address this limitation in the discussion section in the revised manuscript.
>
> Comment 2: Since the authors have a large-scale dataset, a much better approach could be training 3D CNN using patient-level annotation only and compare it, as a baseline, with the proposed method.
> Response 2: We did not compare with a 3D CNN approach because a recent study has shown that using bidirectional Conv-LSTM to scan through a series of consecutive slices of an imaging volume and output as a final set of features characterizing overall disease presence has outperformed a 3D CNN (with equivalent number of parameters for fair comparison) and multiple instance learning (Braman et al., 2018) in detection of emphysema. As PE is actually much smaller than the emphysematous portion of a lung, we expect the advantage of using bidirectional Conv-LSTM over 3D CNN will be even larger.
>
> Comment 3: It is mentioned in the paper that “The attention map is then normalized by its maximum value to range between 0 and 1”. This normalization should make the attention map similar for both negative and positive samples which should hurt the final prediction.
> Response 3: In our actual implementation, the attention map was only normalized if it contains non-zero values. For negative samples, the attention maps were most likely to be all zeroes, and normalization was not performed. Moreover, attention map normalization was only performed during the training in Stage I. The final patient level prediction depends on the output of the last convolutional layer of the trained ResNet in stage I, and thus does not rely on the attention map normalization. Therefore, this normalization would not hurt the final prediction. We will clarify this in the revised manuscript. Nonetheless, we agree that some false positive predictions can also have positive attention maps, so training the network without attention map normalization is surely worth investigating in the future.
>
> Comment 4: Using 2.5mm as slice thickness is probably too thick for detecting subsegment PE.
> Response 4: We agree with the reviewer that 2.5mm slice spacing might be too thick for detecting some sub-segmental PEs. However, these small embolisms are less actionable and less dangerous than larger ones. In addition, we are interested in analyzing all contrast-enhanced CT images, not only CTPA ones that usually have a small slice spacing. In many cases the original image is going to have a slice thickness larger than 2.5 mm. As an example, our test set images have slice spacings ranging from 0.8 to 5mm, with a median of 2 mm. Finally, we used 2.5mm to reduce training time and required GPU memory. The model can be re-trained with smaller slice thicknesses without any change to the architecture or methodology.

---

> ### Author Response · Authors · 2020-04-03
> **Adding the result of 3D CNN for another baseline method comparison**
>
> To compare to a baseline model with 3D CNN, we composed a model that starts with an I3D model (3D CNN pretrained on action recognition) followed by Conv LSTM layers and dense layers as the classifier. This network structure has demonstrated success in acute aortic syndrome detection, in a soon to be published paper.
>
> The network was trained only on the patient-labeled data, and resulted in an AUC of 0.787 on our test set, which is not as good as the AUC of 0.812 by our proposed method.
>
> We will add details about the baseline model and comparison to our model in the final paper.

---

### Meta-Review · Area_Chair1 · 2020-04-05
**MetaReview of Paper42 by AreaChair1**

**Rating:** 4
**Recommendation For Accepted Papers:** Oral

**Metareview:**

Based on the reviewer comments which are all positive about the methodology soundness, clinical value, large-scale dataset employed, and good presentation of the paper, I would like to recommend acceptance. The authors should address the issues raised by reviewers regarding comparison with state-of-the-art methods in the experiments.

**Paper Type:**

both

**Special Issue:**

no

---

### Decision · Program_Chairs · 2020-04-11

Accept